# Effect of renin-angiotensin-aldosterone system inhibitors on Covid-19 patients in Korea

**Jungchan Park** [1‡], **Seung-Hwa Lee** [2☯‡*], **Seng Chan You** [3], **Jinseob Kim** [4], **Kwangmo Yang** [5☯*]

**1** Department of Anesthesiology and Pain Medicine, Samsung Medical Center, Sungkyunkwan University School of Medicine, Seoul, Korea, **2** Division of Cardiology, Department of Medicine, Heart Vascular Stroke Institute, Samsung Medical Center, Sungkyunkwan University School of Medicine, Seoul, Korea, **3** Department of Biomedical Sciences, Ajou University Graduate School of Medicine, Suwon, Korea, **4** Department of Epidemiology, School of Public Health, Seoul National University, Seoul, Korea, **5** Center for Health Promotion, Samsung Medical Center, Sungkyunkwan University School of Medicine, Seoul, Korea

☯ These authors contributed equally to this work.
‡ These authors share first authorship on this work and contributed equally to this work
* shuaaa.lee@samsung.com (S-HL); kmhi.yang@samsung.com (KY)

**Data Availability Statement:** We used de-identified data based on the insurance benefit claims sent to the Health Insurance Review and Assessment Service of Korea (HIRA). This data set is comprised of all patients who were tested for

## Abstract

### Background

The effect of renin-angiotensin-aldosterone system (RAAS) inhibitors in coronavirus disease 19 (Covid-19) patients has not been fully investigated. We evaluated the association between RAAS inhibitor use and outcomes of Covid-19.

### Methods

This study was a retrospective observational cohort study that used data based on insurance benefit claims sent to the Health Insurance Review and Assessment Service of Korea by May 15, 2020. These claims comprised all Covid-19 tested cases and the history of medical service use in these patients for the past five years. The primary outcome was all-cause mortality, and the rate of ventilator care was compared between the groups.

### Results

From a total of 7,590 patients diagnosed with Covid-19, two distinct cohorts were generated based on RAAS inhibitors prescribed within 6 months before Covid-19 diagnosis. A total of 1,111 patients was prescribed RAAS inhibitors, and 794 patients were prescribed antihypertensive drugs, excluding RAAS inhibitors. In propensity-score matched analysis, 666 pairs of data set were generated, and all-cause mortality of the RAAS inhibitor group showed no significant difference compared with the non-RAAS inhibitor group (14.6% vs. 11.1%; hazard ratio [HR], 0.79; 95% confidence interval [CI], 0.54–1.15; p = 0.22). The rate of ventilator care was not significantly different between the two groups (4.4% vs. 4.1%; HR, 1.04; 95% CI, 0.60–1.79; p = 0.89).

Covid-19 in Korea until May 15, 2020, including the history of medical service used by these patients for the past five years. The data are shared in the form of the Observational Medical Outcome Partnership Common Data Model (OMOP-CDM). Observational Health Data Sciences and Informatics (OHDSI) analysis tools are built into the ATLAS interactive analysis platform and the OHDSI Methods Library R packages. OHDSI's open-source software is publicly available on the GitHub repository (https://github.com/OHDSI/). In addition, concept sets used to define baseline characteristics and study outcomes are also available (https://github.com/OHDSI/Covid-19/).

**Funding:** This research was funded by the Ministry of Health and Welfare, Korea (grant number: HI19C0811).

**Competing interests:** The authors declare no conflicts of interest.

## Conclusions

RAAS inhibitor treatment did not appear to increase the mortality of Covid-19 patients compared with other antihypertensive drugs, suggesting that they may be safely continued in Covid-19 patients.

## Introduction

In December 2019, a major outbreak of severe acute respiratory syndrome coronavirus 2 (SARS-Cov-2) was first reported in Wuhan City, China. The World Health Organization characterized the coronavirus disease 19 (Covid-19) as a pandemic on March 11, 2020, and Covid-19 has become a global threatening disease with more than 6,000,000 confirmed cases worldwide as of June 2020 [1]. In Korea, more than 11,629 cases of Covid-19 were diagnosed and 273 deaths were reported through June 2020, and the government of Korea decided to share the world's first de-identified Covid-19 nationwide patient data collected from the Korean National Health Insurance System.

The initial epidemiologic report on Covid-19 indicated that mortality was much higher in patients with cardiovascular disease [2]. Concerns arose on use of renin-angiotensin-aldosterone system (RAAS) inhibitors, because SARS-Cov-2 is known to interact with RAAS through angiotensin-converting enzyme 2 (ACE2) as a receptor [3]. The ACE2 upregulated by RAAS inhibitors could theoretically initiate SARS-Cov-2 infection and aggravate Covid-19 virulence [3]. In contrast, mechanistic evidence from other coronaviruses suggest that downregulation of ACE2 in infected patients leads to acute lung injury, and use of RAAS inhibitors could mitigate this effect by upregulating ACE2 [4–6]. Based on these laboratory findings and considering that the indications of RAAS inhibitors include high-risk patients, it is important to define guidelines on whether to continue RAAS inhibitors in Covid-19 patients rather than switch to other antihypertensive drugs [3,7–9]. However, as the clinical data for these patients are limited, there could be serious health consequences if RAAS inhibitors are associated with Covid-19 mortality. Although a recommendation against suspension of RAAS inhibitors was made by the statement from the American and European Cardiology Societies [10], it was not based on clinical evidence, and the absence of clinical evidence supporting the safety of RAAS inhibitors in Covid-19 patients leaves clinicians with no choice but to follow the old principle of "*primum non nocere (first, do no harm)."* Therefore, we used de-identified Covid-19 nationwide data from Korea to evaluate the association between RAAS inhibitor use and severe Covid-19 induced outcomes. Our results may provide evidence for guidance on use of RAAS inhibitors in Covid-19 patients.

## Materials and methods

### Data curation

This was a retrospective observational cohort study conducted in accordance with the principles of the Declaration of Helsinki. The Institutional Review Board of Samsung Medical Center granted a waiver of approval and informed consent for this study (SMC 2020-04-009) since we used de-identified data based on the insurance benefit claims sent to the Health Insurance Review and Assessment Service of Korea (HIRA). This data set is comprised of all patients who were tested for Covid-19 in Korea until May 15, 2020, including the history of medical

service used by these patients for the past five years. The data are shared in the form of the Observational Medical Outcome Partnership Common Data Model (OMOP-CDM) [11,12].

### Cohort definition and outcomes

The target cohort was generated by selecting hypertensive patients with RAAS inhibitor prescriptions, including angiotensin converting enzyme inhibitor (ACEi) or angiotensin II receptor blocker (ARB), within 6 months prior to Covid-19 diagnosis. For the comparator cohort, we selected patients with antihypertensive drug prescriptions other than RAAS inhibitors within 6 months prior to Covid-19 diagnosis. The target cohort was a RAAS inhibitor group, and the comparator cohort was a non-RAAS inhibitor group. We extracted the incidence of each baseline characteristic without an exact number of patients to protect sensitive personal information and maintain a de-identified form of the data. The primary outcome was all-cause mortality. To compare the incidence of ventilator care as the secondary outcome, both cohorts were re-generated after excluding patients on ventilator care after antihypertensive drug prescriptions and before Covid-19 diagnosis.

### Statistical analysis

Observational Health Data Sciences and Informatics (OHDSI) analysis tools are built into the ATLAS interactive analysis platform and the OHDSI Methods Library R packages. OHDSI's open-source software is publicly available on the GitHub repository (https://github.com/OHDSI/). In addition, concept sets used to define baseline characteristics and study outcomes are also available (https://github.com/OHDSI/Covid-19/). ATLAS ver. 2.7.2 was used herein. As OHDSI CDM does not provide exact numbers of patients for each covariate, we presented incidences of baseline characteristics. To minimize the effects of potential confounding factors and selection bias, we used large-scale propensity score matching and generated a matched population from the cohorts. Cox regression analysis was used to compare outcomes according to RAAS inhibitor use. Kaplan-Meier estimates were used to construct survival curves after propensity-score stratification and compared with the log-rank test. All tests were two-tailed, and $p < 0.05$ was considered statistically significant.

### Results

Data from the insurance benefit claims sent to HIRA until May 15, 2020 indicated that a total of 7,590 patients was diagnosed with Covid-19. Among these patients, the target cohort was generated by selecting 1,111 patients prescribed RAAS inhibitors within 6 months before diagnosis, and the comparator cohort was generated by selecting 794 patients prescribed other antihypertensive drugs in the same time frame (Fig 1). Baseline characteristics are shown in Table 1. The median follow-up duration was 68 days (interquartile range 60–79) in the RAAS inhibitor group and 68 days (interquartile range 58–80) in the non-RAAS inhibitor group. A total of 666 pairs of well-balanced groups was generated after propensity score matching (Table 1 and Fig 2). In the propensity-score matched analysis, all-cause mortality of the RAAS inhibitor group showed no significant difference compared with that of the non-RAAS inhibitor group (14.6% vs. 11.1%; hazard ratio [HR], 0.79; 95% confidence interval [CI], 0.54–1.15; p = 0.22) (Table 2 and Fig 3).

For ventilator care comparison, 20 patients that needed ventilator care between antihypertensive drug prescription and Covid-19 diagnosis were excluded (S1 Fig). The target cohort consisted of 1,098 patients on RAAS inhibitor treatment, and the comparator cohort consisted of 787 patients with other antihypertensive drug treatment (S1 Table). After propensity score matching, a total of 660 pairs was generated, and we found no significant imbalance between

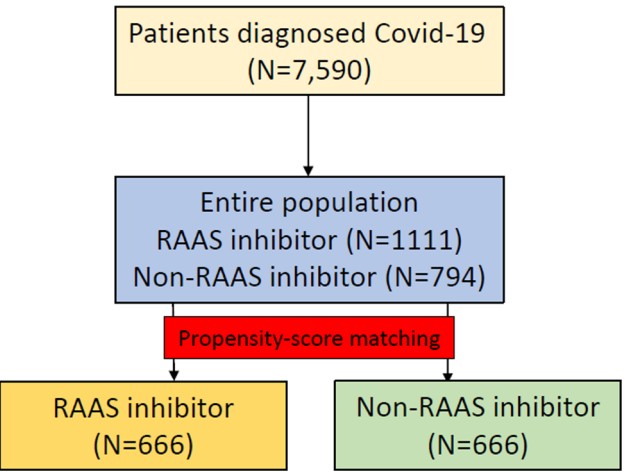

**Fig 1. The flowchart of patients.**

the groups (S1 Table and S2 Fig). The incidence of ventilator care also showed no difference (4.4% vs. 4.1%; HR, 1.04; 95% CI, 0.60–1.79; p = 0.89) (Table 2 and S3 Fig).

## Discussion

In the current study, use of RAAS inhibitors in Covid-19 patients did not appear to be associated with higher mortality compared with that of other antihypertensive drugs. The results of our study are in agreement with recently reported studies and the current recommendations [3,7–9], and add evidence to that RAAS inhibitor treatment should be continued in Covid-19 patients.

Based on a finding from the first major outbreak of the severe acute respiratory syndrome coronavirus (SARS-Cov) in Hong Kong in 2003 that ACE2 acts as a functional receptor for coronavirus [13,14], Sommerstein and Gra¨ni presented a hypothesis that ACEi could act as a potential risk factor of Covid-19, and upregulation of ACE2 could cause fatal outcomes [15]. ACE2 was found to be the receptor-binding site for the spike protein of Covid-19 and well as in SARS-Cov, and concerns were raised on use of RAAS inhibitors in Covid-19 patients. Some authors argued that patients should discontinue RAAS inhibitors, even temporarily, given the current pandemic of Covid-19 [3,16,17]. Although the evidence for this hypothesis is insufficient and is mostly derived from in vitro studies, clinical data to refuse this hypothesis are not available. Therefore, replacing RAAS inhibitors with other antihypertensive drugs in Covid-19 patients remains controversial [18].

Some recently published commentaries have recommended against suspension or withdrawal of RAAS inhibitors [3,7–9] based on evidence from several animal and experimental models [19–21]. Several observational analysis also showed that there was no significant association between RAAS inhibitor treatment and outcomes of Covid-19 [22,23], but these studies used single-center data with a small number of patients. In this study, we reinforced the recommendation against withdrawing RAAS inhibitors by presenting real-world data of a dedicated nationwide Covid-19 patient registry. Although differences in the early stage of Covid-19 according to the use of RAAS inhibitor was unclear, we demonstrated that it was not associated with increased mortality of Covid-19 compared with other antihypertensive drugs. The previously reported effect of RAAS inhibitors on ACE2 level and activity in humans is unclear [19,20,24]. The results of studies conducted for SARS-Cov, if generalizable to Covid-19, suggest that the effect of ARB may paradoxically be protective against Covid-19 [4]. The

**Table 1. Baseline characteristics.**

| | Before propensity score adjustment | | | After propensity score adjustment | | |
|---|---|---|---|---|---|---|
| | RAAS inhibitor (N = 1,111) | Non-RAAS inhibitor (N = 794) | SMD | RAAS inhibitor (N = 666) | Non-RAAS inhibitor (N = 666) | SMD |
| **Age group** | | | | | | |
| 15–19 | 0.1 | 0.6 | -0.09 | 0.2 | 0.6 | -0.07 |
| 20–24 | 0.5 | 2.1 | -0.15 | 0.6 | 2 | -0.12 |
| 25–29 | 0.9 | 3.4 | -0.17 | 1.2 | 2.7 | -0.11 |
| 30–34 | 1 | 2.1 | -0.09 | 1.2 | 2 | -0.06 |
| 35–39 | 1.4 | 1.4 | 0 | 0.9 | 1.1 | -0.01 |
| 40–44 | 1.9 | 3.3 | -0.09 | 2.3 | 2.9 | -0.04 |
| 45–49 | 6.2 | 6.3 | 0 | 6.9 | 6.2 | 0.03 |
| 50–54 | 10.1 | 8.9 | 0.04 | 9.9 | 8.3 | 0.06 |
| 55–59 | 14.5 | 12 | 0.07 | 15.3 | 11.7 | 0.11 |
| 60–64 | 16.1 | 11.7 | 0.13 | 15.3 | 11.7 | 0.11 |
| 65–69 | 12.1 | 11.8 | 0.01 | 10.4 | 13.2 | -0.09 |
| 70–74 | 10.1 | 10.5 | -0.01 | 10.5 | 11 | -0.01 |
| 75–79 | 10.8 | 11.2 | -0.08 | 9.8 | 11.8 | -0.08 |
| 80–84 | 7.7 | 6.7 | 0.04 | 7.8 | 6.9 | 0.04 |
| 85–89 | 4.8 | 5 | -0.01 | 5.1 | 5 | 0.01 |
| 90–94 | 1.5 | 2.4 | -0.06 | 2.1 | 2.3 | -0.01 |
| 95–99 | 0.3 | 0.6 | -0.05 | 0.5 | 0.6 | -0.02 |
| **Sex: Female** | 55.9 | 55.4 | 0.01 | 55.4 | 54.5 | 0.02 |
| **Medical history** | | | | | | |
| Acute respiratory disease | 74.2 | 70.8 | 0.08 | 71.9 | 70.9 | 0.02 |
| Chronic liver disease | 9.6 | 8.6 | 0.04 | 8.7 | 9.8 | -0.04 |
| Chronic obstructive lung disease | 3.8 | 4.7 | -0.04 | 3.8 | 4.5 | -0.04 |
| Dementia | 11.7 | 16.9 | -0.15 | 14 | 16.4 | -0.07 |
| Depressive disorder | 19 | 27.2 | -0.2 | 23.7 | 21.9 | 0.04 |
| Diabetes mellitus | 41.6 | 28.2 | 0.28 | 30.2 | 32.6 | -0.05 |
| Gastroesophageal reflux disease | 44.8 | 44.7 | 0 | 42 | 45.8 | -0.08 |
| Gastrointestinal hemorrhage | 3.3 | 3.8 | -0.02 | 3.5 | 3.9 | -0.02 |
| Hyperlipidemia | 70.7 | 53.7 | 0.36 | 56.5 | 62.5 | -0.12 |
| Lesion of liver | 4 | 3 | 0.05 | 3.2 | 3.3 | -0.01 |
| Obesity | 0.3 | 0.1 | 0.03 | 0.5 | 0.2 | 0.06 |
| Osteoarthritis | 26 | 26.4 | -0.01 | 24 | 27.5 | -0.08 |
| Pneumonia | 51.8 | 49.5 | 0.05 | 49.8 | 50.9 | -0.02 |
| Psoriasis | 1.8 | 0.8 | 0.09 | 1.8 | 0.8 | 0.09 |
| Renal impairment | 6.8 | 4.7 | 0.09 | 5.6 | 5.3 | 0.01 |
| Rheumatoid arthritis | 4.7 | 4.3 | 0.02 | 4.4 | 5 | -0.03 |
| Schizophrenia | 3.7 | 6.7 | -0.14 | 5.1 | 5.4 | -0.01 |
| Urinary tract infectious disease | 7.5 | 9.4 | -0.07 | 7.7 | 9 | -0.05 |
| Viral hepatitis C | 0.5 | 1 | -0.05 | 0.3 | 1.1 | -0.09 |
| Visual system disorder | 49.3 | 49.6 | -0.01 | 46.1 | 50.6 | -0.09 |
| **Medical history: Cardiovascular disease** | | | | | | |
| Atrial fibrillation | 3.2 | 3.8 | -0.03 | 3 | 4.1 | -0.06 |
| Cerebrovascular disease | 9.2 | 6.8 | 0.09 | 8.9 | 8 | 0.03 |
| Coronary arteriosclerosis | 0.7 | 1.8 | -0.09 | 0.9 | 2 | -0.09 |
| Heart disease | 33.1 | 31.6 | 0.03 | 32 | 33.5 | -0.03 |

(*Continued*)

**Table 1.** (Continued)

| | Before propensity score adjustment | | | After propensity score adjustment | | |
|---|---|---|---|---|---|---|
| | RAAS inhibitor | Non-RAAS inhibitor | SMD | RAAS inhibitor | Non-RAAS inhibitor | SMD |
| | (N = 1,111) | (N = 794) | | (N = 666) | (N = 666) | |
| Ischemic heart disease | 15.6 | 13.7 | 0.05 | 15.8 | 14.6 | 0.03 |
| Peripheral vascular disease | 17.2 | 15.7 | 0.04 | 15.3 | 17.9 | -0.07 |
| Pulmonary embolism | 21.3 | 17.8 | 0.09 | 21.2 | 19.2 | 0.05 |
| Venous thrombosis | 0.2 | 0.8 | -0.08 | 0.3 | 0.8 | -0.06 |
| **Medical history: Neoplasms** | | | | | | |
| Hematologic neoplasm | 0.5 | 0.6 | -0.03 | 0.3 | 0.5 | -0.03 |
| Malignant lymphoma | 0.1 | 0.3 | -0.04 | 0.2 | 0.3 | -0.03 |
| Malignant neoplastic disease | 7.7 | 8.6 | -0.03 | 7.2 | 8.7 | -0.06 |
| Malignant tumor of breast | 0.6 | 0.4 | 0.04 | 0.5 | 0.3 | 0.03 |
| Malignant tumor of colon | 0.5 | 0.6 | -0.03 | 0.2 | 0.8 | -0.09 |
| Malignant tumor of lung | 0.6 | 0.4 | 0.04 | 0.6 | 0.3 | 0.04 |
| Malignant tumor of urinary bladder | 0.2 | 0.1 | 0.01 | 0.3 | 0.2 | 0.03 |
| Primary malignant neoplasm of prostate | 1.2 | 1.5 | -0.03 | 1.1 | 1.7 | -0.05 |

Data are presented as %.

RAAS, renin-angiotensin-aldosterone system; SMD, standardized mean difference.

underlying mechanism could be related to the possibility that interaction with the coronavirus may lead to ACE2 downregulation. This, in turn, causes excessive production of angiotensin by ACE, whereas lower level of ACE2 allows for conversion to angiotensin (1–7), which is a heptapeptide with vasodilator activity [5]. With ACE2 downregulation, angiotensin-II stimulates AT1 to increase pulmonary vascular permeability, thereby mediating increased lung pathology [6]. Use of RAAS inhibitors may upregulate ACE2 to compensate for this downregulation, and this could cause higher ACE2 expression to protect against acute lung injury in SARS-Cov-2 infected patients rather than to increase the risk of COVID-19.

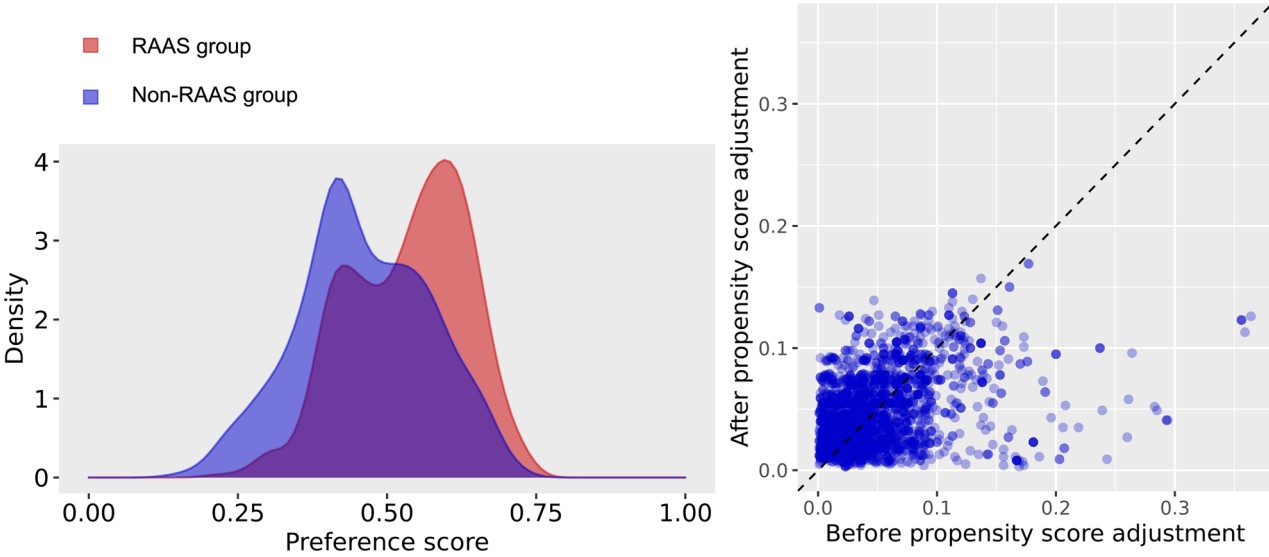

**Fig 2. Balance between the groups before and after propensity score matching.**

**Table 2. Clinical outcomes.**

| | Before propensity-score stratification | | | | After propensity-score stratification | | | |
|---|---|---|---|---|---|---|---|---|
| | RAAS inhibitor | Non-RAAS inhibitor | Unadjusted | p value | RAAS inhibitor | Non-RAAS inhibitor | Adjusted | p value |
| | (N = 1,111) | (N = 794) | HR (95% CI) | | (N = 666) | (N = 666) | HR (95% CI) | |
| All-cause mortality | 97 (8.7) | 74 (9.3) | 0.96 (0.71–1.30) | 0.79 | 97 (14.6) | 74 (11.1) | 0.79 (0.54–1.15) | 0.22 |
| | RAAS inhibitor | Non-RAAS inhibitor | Unadjusted | p value | RAAS inhibitor | Non-RAAS inhibitor | Adjusted | p value |
| | (N = 1098) | (N = 787) | HR (95% CI) | | (N = 660) | (N = 660) | HR (95% CI) | |
| Ventilator care | 54 (4.9) | 31 (3.9) | 1.23 (0.80–1.93) | 0.36 | 29 (4.4) | 27 (4.1) | 1.04 (0.60–1.79) | 0.89 |

Data are presented as %.

RAAS, renin-angiotensin-aldosterone system; HR, hazard ratio; CI, confidence interval.

Another issue that should be considered is the beneficial effect of RAAS inhibitors compared with other antihypertensive drugs in patients with heart disease [25]. RAAS inhibitors are antihypertensive drugs that should be used in patients with heart failure and for secondary prevention after acute myocardial infarction [25,26]. Among comorbidities of Covid-19, patients with cardiovascular disease have shown higher fatality rate [2]. With the systemic inflammatory response and immune system disorders that can occur during disease progression, Covid-19

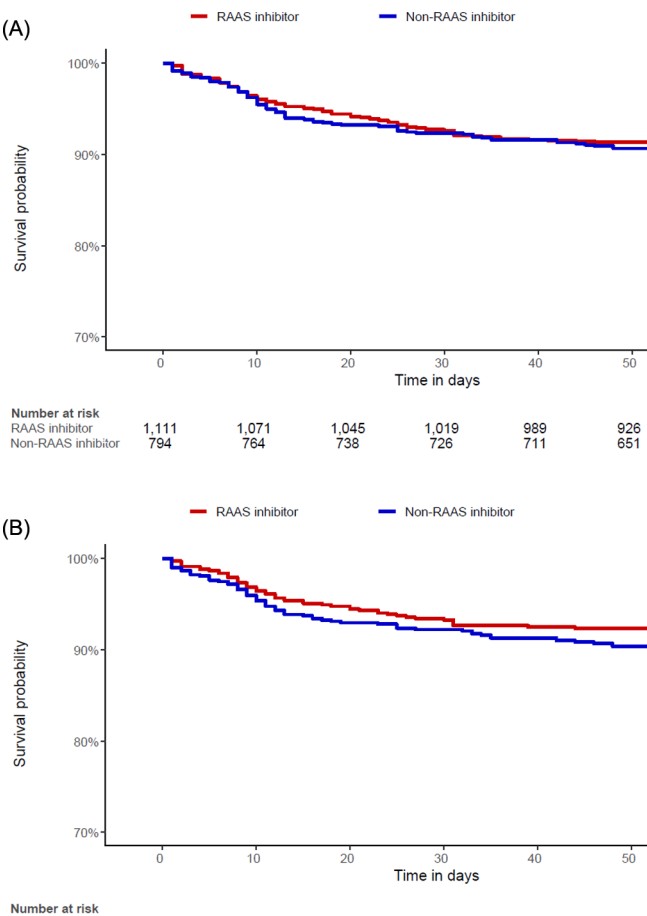

**Fig 3.** Kaplan-Meier curves for mortality in the (A) entire population and (B) propensity score matched population.

patients can be more vulnerable to cardiovascular disorders such as myocardial injury [18]. Myocardial injury associated with SARS-CoV-2 occurred in 5 of the first 41 patients diagnosed in Wuhan [27], and Covid-19 patients with acute myocardial injury showed higher mortality than other patients [28]. Taken these together, the cardioprotective effect of RAAS inhibitors could even be helpful for outcomes of Covid-19 patients with cardiovascular disease, but this is beyond the scope of the present study and needs further investigation.

A previous report presented the difference between ACEi and ARB in the association with increased intestinal ACE messenger RNA levels and found that it was associated with ACEi but not ARB [29,30]. However, we could not conduct separate analyses based on the types of RAAS inhibitors in this study due to the enormously higher rate of ARB use. ARB has previously been reported to have a higher rate of use than ACEi in Korea [31], because the side effects of ACEi such as cough or angioedema are relatively more frequent in the Asian population. The ongoing study of "Losartan for Patients with COVID-19 Requiring Hospitalization (NCT04312009)" may provide evidence for the use of ARB, separate from ACEi.

The clinical implications of this study are that we added evidence that supports continued use of RAAS inhibitors in Covid-19 patients. RAAS inhibitors are approved for heart failure, diabetic nephropathy, and secondary prevention after acute myocardial infarction, but not all other antihypertensive drugs cover these indications [25]. Patients on RAAS inhibitor treatment are likely to be more fragile, and replacing RAAS inhibitors with other antihypertensive drugs may cause increased risk of adverse cardiovascular events. Indeed, spironolactone has been proposed as an alternative of RAAS inhibitors and even as a potential therapy for Covid-19 [32]. A number of studies on use of RAAS inhibitors in Covid-19 patients has been published and based on the data from the beginning of the outbreak, although some of them were recently retracted because of reliability and accuracy issues of the data. The result of the present study was based on reliable nationwide data from the government of Korea and supports the current recommendations that RAAS inhibitors should be continued in Covid-19 patients.

The results of this study should be interpreted with consideration of the following limitations. First, this was a retrospective study. Despite our efforts to adjust all confounding factors by propensity score matching analysis, some of the covariates were not well balanced in the propensity score matched population, and unmeasured factors might have affected the results. Second, owing to the nature of the database that retrieved the information from insurance issued claims, clinical presentation, symptoms, and hospital course could not be evaluated. Furthermore, a need for hospitalization or a radiologic evidence of lung injury represents a difference in response to the first stage, but these parameters could not be curated. Third, whether the patients actually continued or stopped taking RAAS inhibitors after diagnosis of Covid-19 could not be accurately evaluated. In addition, a comparison between the types of RAAS inhibitors (ACEi vs. ARB) was not performed. Lastly, the results of the current study are derived from a cohort of Korea; hence, the impact of ethnicity cannot be analyzed and needs further evaluation. Despite these limitations, this study provides the first real-world evidence on use of RAAS inhibitors in Covid-19 patients and valuable information for patient treatment during this pandemic.

## Conclusion

In this study, RAAS inhibitor treatment did not appear to increase the mortality of Covid-19 patients compared with other antihypertensive drugs. Based on the results of the current study and previous recommendations, RAAS inhibitors may safely be continued in Covid-19 patients.

## Supporting information

**S1 Fig.**
(TIF)

**S2 Fig.**
(TIF)

**S3 Fig.**
(TIF)

**S1 Table. Baseline characteristics of cohorts for ventilator care comparison.**
(DOCX)

## Acknowledgments

The authors appreciate healthcare professionals dedicated to treating Covid-19 patients in Korea, and the Ministry of Health and Welfare and the Health Insurance Review & Assessment Service of Korea for sharing valuable national health insurance claims data in a prompt manner.

## Author Contributions

**Conceptualization:** Seung-Hwa Lee.

**Data curation:** Jungchan Park, Seng Chan You, Jinseob Kim.

**Formal analysis:** Seng Chan You, Jinseob Kim.

**Funding acquisition:** Seung-Hwa Lee, Kwangmo Yang.

**Investigation:** Jungchan Park, Seung-Hwa Lee, Kwangmo Yang.

**Software:** Jungchan Park, Seung-Hwa Lee, Seng Chan You.

**Supervision:** Kwangmo Yang.

**Validation:** Seng Chan You.

**Visualization:** Kwangmo Yang.

**Writing – original draft:** Jungchan Park, Seung-Hwa Lee.

**Writing – review & editing:** Seng Chan You, Jinseob Kim, Kwangmo Yang.

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
