## [Decision Letter · Decision Letter 0]

21 Oct 2020

PONE-D-20-27008

Effect of Renin-Angiotensin-Aldosterone System Inhibitors on Covid-19 Patients in Korea

PLOS ONE

Dear Dr. Lee,

Thank you for submitting your manuscript to PLOS ONE. After careful consideration, we feel that it has merit but does not fully meet PLOS ONE’s publication criteria as it currently stands. Therefore, we invite you to submit a revised version of the manuscript that addresses the points raised during the review process.

We look forward to receiving your revised manuscript.

Kind regards,

Giuseppe Vergaro, M.D..Ph.D.

Academic Editor

PLOS ONE

Journal Requirements:

2. Data availability. Please note that PLOS journals require authors to make all data underlying the findings described in their manuscript fully available without restriction, with rare exception. Unfortunately, your statement that "some restrictions will apply", it is not in accordance with PLOS data availability policy. PLOS requires that a “minimal data set” is shared, defined as the data set used to reach the conclusions drawn in the manuscript with related metadata and methods, and any additional data required to replicate the reported study findings in their entirety. Authors do not need to submit their entire data set if only a portion of the data were used in the reported study. Also, authors do not need to submit the raw data collected during an investigation if the standard in the field is to share data that have been processed. Please submit the following data: The values behind the means, standard deviations and other measures reported; The values used to build graphs; The points extracted from images for analysis.” http://journals.plos.org/plosone/s/data-availability#loc-faqs-for-data-policy. If you are unable to share the data, this may result in manuscript rejection.

Reviewers' comments:

Reviewer's Responses to Questions

**Comments to the Author**

1. Is the manuscript technically sound, and do the data support the conclusions?

Reviewer #1: Yes

Reviewer #2: Yes

2. Has the statistical analysis been performed appropriately and rigorously? 

Reviewer #1: Yes

Reviewer #2: Yes

3. Have the authors made all data underlying the findings in their manuscript fully available?

Reviewer #1: No

Reviewer #2: No

4. Is the manuscript presented in an intelligible fashion and written in standard English?

Reviewer #1: No

Reviewer #2: Yes

5. Review Comments to the Author

Reviewer #1: Park et al. performed a retrospective chart analysis on the impact of previous use of RAAS inhibitors on COVID-19 related clinical outcomes. Authors reported no difference between the group of patients treated with RAAS-inhibitors when compared to controls. The study seems very interesting and of relevant clinical interest, but probably needs language check, eventually performed by a native speakers.

1. Abstract conclusion should be more cautious since this is a retrospective analysis that may generate robust hypothesis but cannot gather general conclusion. Please tone the sentence down

2. line 170. the sentence “that use of RAAS inhibitors in Covid-19 patients was not associated with a greater incidence of mortality compared with that of other antihypertensive drugs” should be rephrased in a more proper way.

3. At line 189 the authors speculate that interaction with the coronavirus leads to ACE2 down regulation. As assessed by Michele M. Ciulla (Ciulla, M.M.; 2020). However, there are no scientific evidences that SARS-CoV-2 downregulates ACE2. Is that only authors’ speculation? Did the authors have any data about the expression regulation of ACE? Otherwise I would suggest also in this case to tone down the sentence

4. Line 195. Authors claim that RAAS inhibitors induce a higher expression level of ACE2. Again, is there any data that can support this statement?

5. Line 201. The sentence “Cardiovascular disease in Covid-19 patients showed higher fatality rates than other comorbidities such as diabetes or cardiovascular disease” is not clear. Please rephrase

6. At line 207 “Therefore, the cardioprotective effect of RAAS inhibitors may have decreased the mortality in Covid-19 patients”. This conclusion does not appear to be a result of the study. The authors so far stated that they evaluated the incidence of mortality by comparing Covid-19 positive patients treated with either RAAS inhibitors or other types of antihypertensive therapies. Do the authors have any data that can validate their statement?

7. Line 210 “A previous report presented the difference between ACEi and ARB in the association with increased intestinal messenger RNA levels and found that it was associated with ACEi but not ARB”. This sentence is of not clear understanding what type of mRNA is up regulated?.

8. Line 243 “In Covid-19 patients, RAAS inhibitor treatment was not associated with mortality”. According to the results presented by the authors this conclusion is wrong. RAAS inhibitor treatment was not associated with higher mortality compared to other antihypertensive therapies.

Reviewer #2: Introduction:

The introduction follows a logical sequence and was written concisely. This is of great importance.

Perhaps, I would recommend authors to include the statement from the American and European Cardiology Societies not recommending the suspension in early April, although this recommendation against the suspension was not based in any study confirming the safety of these classes for COVID-19.

I also highly recommend considering to bring the papers from the author below, either in the introduction or in the discussion.

This is an author that has interestingly progressed this thoughts from the perception that ACEi and ARB would lead to increased risk, and therefore proposing spironolactone as an alternative, to further studies no longer recommending to replace ACEi and ARB, and his last article, a very interesting one, where he even proposes ACEi and ARB as potential therapies for COVID-19 if acutely used. Please find the sequence below:

April/2020:

Cadegiani FA. Can spironolactone be used to prevent COVID-19-induced acute respiratory distress syndrome in patients with hypertension? Am J Physiol Endocrinol Metab. 2020 May 1;318(5):E587-E588. doi: 10.1152/ajpendo.00136.2020.

July/2020 (published online in July):

Cadegiani FA, Goren A, Wambier CG. Spironolactone may provide protection from SARS-CoV-2: Targeting androgens, angiotensin converting enzyme 2 (ACE2), and renin-angiotensin-aldosterone system (RAAS). Med Hypotheses. 2020 Oct;143:110112. doi: 10.1016/j.mehy.2020.110112.

Cadegiani FA, Wambier CG, Goren A. Spironolactone: An Anti-androgenic and Anti-hypertensive Drug That May Provide Protection Against the Novel Coronavirus (SARS-CoV-2) Induced Acute Respiratory Distress Syndrome (ARDS) in COVID-19. Front Med (Lausanne). 2020 Jul 28;7:453. doi: 10.3389/fmed.2020.00453.

September/2020:

Cadegiani FA. Repurposing existing drugs for COVID-19: an endocrinology perspective. BMC Endocr Disord. 2020 Sep 29;20(1):149. doi: 10.1186/s12902-020-00626-0.

When only hospitalized patients are considered for the analysis, it is unexpected to detect differences between users and non-users of ECAi and ARB, because the potential damage caused by these drug classes is based on the enhanced viral cell entry due to increased attached ACE2 availability, which only occurs in the first stage.

Among hospitalized patients, that are in common all in the second or third stage, ECAi and ARB should no longer influence outcomes negatively.

Hence, to detect differences overall populations infected by COVID-19 would be more appropriate, and outcomes including hospitalization rate, WHO COVID Ordinal Outcomes, among others, would detect differences more accurately, if there is any.

Lines 82-83: “Therefore, we used de-identified Covid-82 19 nationwide data from Korea to evaluate the association between RAAS inhibitor use and Covid-19 outcomes.”

- If authors only considered mechanical ventilator and mortality, the objective was to detect severe COVID-19 induced outcomes, not all outcomes.

Materials and methods:

- Please specify whether the data was extracted from all patients diagnosed with COVID-19, or only those who got hospitalized. In this matter, there are two options that need to be adjusted, regardless:

1. In case all positive COVID-19 using ACEi/BRA versus other antihypertensives were considered, regardless of needing hospitalization: Inclusion of additional outcomes, including hospitalization, presence of lung injury through chest CT scan, and other parameters that better detect differences in responses to the first stage, when ACEi/BRA could theoretically play as an aggravating factor, would provide more sensitive data, and unveil early differences. However, in this case, it will be important to mention that regardless of differences in these additional parameters, disease course in terms of severity and death were not different.

2. In case only hospitalized patients were included for the analysis, this must be highlighted: “We compared patients that were hospitalized for COVID-19”.

One of the two options above should be addressed.

Results

I must congratulate the authors for the number of variables they adjusted for. Very few works have worked so well on this.

Discussion

I would bring to the discussion that whether there are differences in the early stage of COVID-19 between ACEi/ARB users and non-users is unclear, but regardless of whether these differences exist, they do not seem to impact the need of mechanical ventilation and death.

Again, please read the articles that I mentioned in the introduction. They may be helpful to increase the level of this already great paper.

6. PLOS authors have the option to publish the peer review history of their article (what does this mean?). If published, this will include your full peer review and any attached files.

Reviewer #1: **Yes: **Dr. Alberto M. Marra, MD, PhD

Reviewer #2: No

---

## [Author Response · Author response to Decision Letter 0]

24 Nov 2020

Response to the Reviewers

Reviewer #1.

Park et al. performed a retrospective chart analysis on the impact of previous use of RAAS inhibitors on COVID-19 related clinical outcomes. Authors reported no difference between the group of patients treated with RAAS-inhibitors when compared to controls. The study seems very interesting and of relevant clinical interest, but probably needs language check, eventually performed by a native speakers.

>> Response: We thank you for your kind comments. Following your recommendation, we had our script reviewed by a native professional editor.

 1. Abstract conclusion should be more cautious since this is a retrospective analysis that may generate robust hypothesis but cannot gather general conclusion. Please tone the sentence down

>> Response: We understand your concern. Following your recommendation, we changed our conclusion as below.

“RAAS inhibitor treatment did not appear to increase the mortality of Covid-19 patients compared with other antihypertensive drugs, suggesting that they may be safely continued in Covid-19 patients.” (Line 55)

 2. line 170. the sentence “that use of RAAS inhibitors in Covid-19 patients was not associated with a greater incidence of mortality compared with that of other antihypertensive drugs” should be rephrased in a more proper way.

>> Response: Following your recommendation, we changed it as below.

“In the current study, use of RAAS inhibitors in Covid-19 patients did not appear to be associated with higher mortality compared with that of other antihypertensive drugs.” (Line 164)

 3. At line 189 the authors speculate that interaction with the coronavirus leads to ACE2 down regulation. As assessed by Michele M. Ciulla (Ciulla, M.M.; 2020). However, there are no scientific evidences that SARS-CoV-2 downregulates ACE2. Is that only authors’ speculation? Did the authors have any data about the expression regulation of ACE? Otherwise I would suggest also in this case to tone down the sentence

>> Response: Following your recommendation, we toned it down as below.

“The underlying mechanism could be related to the possibility that interaction with the coronavirus may lead to ACE2 downregulation.” (Line 192)

 4. Line 195. Authors claim that RAAS inhibitors induce a higher expression level of ACE2. Again, is there any data that can support this statement?

>> Response: Following your recommendation, we toned it down as below.

“Use of RAAS inhibitors may upregulate ACE2 to compensate for this downregulation, and this could cause higher ACE2 expression to protect against acute lung injury in SARS-Cov-2 infected patients rather than to increase the risk of COVID-19.” (Line 197)

 5. Line 201. The sentence “Cardiovascular disease in Covid-19 patients showed higher fatality rates than other comorbidities such as diabetes or cardiovascular disease” is not clear. Please rephrase

>> Response: Following your recommendation, we changed it as below.

“Among comorbidities of Covid-19, patients with cardiovascular disease have shown higher fatality rate [2].” (Line 204)

6. At line 207 “Therefore, the cardioprotective effect of RAAS inhibitors may have decreased the mortality in Covid-19 patients”. This conclusion does not appear to be a result of the study. The authors so far stated that they evaluated the incidence of mortality by comparing Covid-19 positive patients treated with either RAAS inhibitors or other types of antihypertensive therapies. Do the authors have any data that can validate their statement?

>> Response: Following your recommendation, we rephrased it as below.

“Taken these together, the cardioprotective effect of RAAS inhibitors could even be helpful for outcomes of Covid-19 patients with cardiovascular disease, but this is beyond the scope of the present study and needs further investigation.” (Line 210)

 7. Line 210 “A previous report presented the difference between ACEi and ARB in the association with increased intestinal messenger RNA levels and found that it was associated with ACEi but not ARB”. This sentence is of not clear understanding what type of mRNA is up regulated?.

>> Response: We agree with the reviewer that this needs more explanation. So, we added a reference and clarified the type of mRNA as below.

“A previous report presented the difference between ACEi and ARB in the association with increased intestinal ACE messenger RNA levels and found that it was associated with ACEi but not ARB [29,30].” (Line 214)

 8. Line 243 “In Covid-19 patients, RAAS inhibitor treatment was not associated with mortality”. According to the results presented by the authors this conclusion is wrong. RAAS inhibitor treatment was not associated with higher mortality compared to other antihypertensive therapies.

>> Response: Following your recommendation, we rephrased it as below. 

“In this study, RAAS inhibitor treatment did not appear to increase the mortality of Covid-19 patients compared with other antihypertensive drugs. Based on the results of the current study and previous recommendations, RAAS inhibitors may safely be continued in Covid-19 patients.” (Line 253)

Reviewer #2.

Introduction:

 The introduction follows a logical sequence and was written concisely. This is of great importance.

 Perhaps, I would recommend authors to include the statement from the American and European Cardiology Societies not recommending the suspension in early April, although this recommendation against the suspension was not based in any study confirming the safety of these classes for COVID-19.

>> Response: We thank you for your kind comments. Following your recommendation, we added as below.

“Although a recommendation against suspension of RAAS inhibitors was made by the statement from the American and European Cardiology Societies [10], it was not based on clinical evidence, and the absence of clinical evidence supporting the safety of RAAS inhibitors in Covid-19 patients leaves clinicians with no choice but to follow the old principle of “primum non nocere (first, do no harm).” (Line 80)

 I also highly recommend considering to bring the papers from the author below, either in the introduction or in the discussion.

 This is an author that has interestingly progressed this thoughts from the perception that ACEi and ARB would lead to increased risk, and therefore proposing spironolactone as an alternative, to further studies no longer recommending to replace ACEi and ARB, and his last article, a very interesting one, where he even proposes ACEi and ARB as potential therapies for COVID-19 if acutely used. Please find the sequence below:

 April/2020:

 Cadegiani FA. Can spironolactone be used to prevent COVID-19-induced acute respiratory distress syndrome in patients with hypertension? Am J Physiol Endocrinol Metab. 2020 May 1;318(5):E587-E588. doi: 10.1152/ajpendo.00136.2020.

 July/2020 (published online in July):

 Cadegiani FA, Goren A, Wambier CG. Spironolactone may provide protection from SARS-CoV-2: Targeting androgens, angiotensin converting enzyme 2 (ACE2), and renin-angiotensin-aldosterone system (RAAS). Med Hypotheses. 2020 Oct;143:110112. doi: 10.1016/j.mehy.2020.110112.

 Cadegiani FA, Wambier CG, Goren A. Spironolactone: An Anti-androgenic and Anti-hypertensive Drug That May Provide Protection Against the Novel Coronavirus (SARS-CoV-2) Induced Acute Respiratory Distress Syndrome (ARDS) in COVID-19. Front Med (Lausanne). 2020 Jul 28;7:453. doi: 10.3389/fmed.2020.00453.

 September/2020:

 Cadegiani FA. Repurposing existing drugs for COVID-19: an endocrinology perspective. BMC Endocr Disord. 2020 Sep 29;20(1):149. doi: 10.1186/s12902-020-00626-0.

>> Response: We agree with the reviewer that mentioning spironolactone as an alternative would be interesting for readers. However, as shown in https://clinicalhypertension.biomedcentral.com/articles/10.1186/s40885-018-0098-0, the prescription rate of spironolactone in Korea is extremely low, so it was not feasible to conduct further study on this. Following your recommendation, we added as below to the Discussino section.

“Indeed, spironolactone has been proposed as an alternative of RAAS inhibitors and even as a potential therapy for Covid-19 [32].” (Line 228)

 When only hospitalized patients are considered for the analysis, it is unexpected to detect differences between users and non-users of ECAi and ARB, because the potential damage caused by these drug classes is based on the enhanced viral cell entry due to increased attached ACE2 availability, which only occurs in the first stage.

 Among hospitalized patients, that are in common all in the second or third stage, ECAi and ARB should no longer influence outcomes negatively.

 Hence, to detect differences overall populations infected by COVID-19 would be more appropriate, and outcomes including hospitalization rate, WHO COVID Ordinal Outcomes, among others, would detect differences more accurately, if there is any.

>> Response: We agree with the reviewer that this needs clarification. In the very beginning of Covid-19 pandemic in Korea, every patients were hospitalized immediately after the diagnosis of Covid-19 regardless of symptom, and starting from April, a quarantine facility for those without symptom were operated. So, our study patients include overall population infected by Covid-19, but unfortunately hospitalization rate could not be extracted from the CDM system. Following your recommendation, we added as below to the Limiation section.

“Second, owing to the nature of the database that retrieved the information from insurance issued claims, clinical presentation, symptoms, and hospital course could not be evaluated. Furthermore, a need for hospitalization or a radiologic evidence of lung injury represents a difference in response to the first stage, but these parameters could not be curated.” (Line 240)

 Lines 82-83: “Therefore, we used de-identified Covid-82 19 nationwide data from Korea to evaluate the association between RAAS inhibitor use and Covid-19 outcomes.”

- If authors only considered mechanical ventilator and mortality, the objective was to detect severe COVID-19 induced outcomes, not all outcomes.

>> Response: Following your recommendation, we changed it as below.

“Therefore, we used de-identified Covid-19 nationwide data from Korea to evaluate the association between RAAS inhibitor use and severe Covid-19 induced outcomes.” (Line 85)

Materials and methods:

 - Please specify whether the data was extracted from all patients diagnosed with COVID-19, or only those who got hospitalized. In this matter, there are two options that need to be adjusted, regardless:

 1. In case all positive COVID-19 using ACEi/BRA versus other antihypertensives were considered, regardless of needing hospitalization: Inclusion of additional outcomes, including hospitalization, presence of lung injury through chest CT scan, and other parameters that better detect differences in responses to the first stage, when ACEi/BRA could theoretically play as an aggravating factor, would provide more sensitive data, and unveil early differences. However, in this case, it will be important to mention that regardless of differences in these additional parameters, disease course in terms of severity and death were not different.

 2. In case only hospitalized patients were included for the analysis, this must be highlighted: “We compared patients that were hospitalized for COVID-19”.

One of the two options above should be addressed.

>> Response: We agree with the reviewer that this needs to be clarified. For our study all the extracted data were among patients who were dignosed with Covid-19. As mentioned above, a quarantine facility for Covid-19 patients without symptom has been operated in Korea, but we could not extract hospitalization rate as well as other parameters that detects the difference in the first stage. Following your recommendation, we added as below to the script.

“Second, owing to the nature of the database that retrieved the information from insurance issued claims, clinical presentation, symptoms, and hospital course could not be evaluated. Furthermore, a need for hospitalization or a radiologic evidence of lung injury represents a difference in response to the first stage, but these parameters could not be curated.” (Line 240)

 Results

 I must congratulate the authors for the number of variables they adjusted for. Very few works have worked so well on this.

 Discussion

 I would bring to the discussion that whether there are differences in the early stage of COVID-19 between ACEi/ARB users and non-users is unclear, but regardless of whether these differences exist, they do not seem to impact the need of mechanical ventilation and death.

>> Response: Following your recommendation, we added as below.

“Although differences in the early stage of Covid-19 according to the use of RAAS inhibitor was unclear, we demonstrated that it was not associated with increased mortality of Covid-19 compared with other antihypertensive drugs.” (Line 186)

Again, please read the articles that I mentioned in the introduction. They may be helpful to increase the level of this already great paper.

>> Response: Following your recommendation, we added as below.

Indeed, spironolactone has been proposed as an alternative of RAAS inhibitors and even as a potential therapy for Covid-19 [32].” (Line 228)

---

## [Decision Letter · Decision Letter 1]

19 Feb 2021

Effect of Renin-Angiotensin-Aldosterone System Inhibitors on Covid-19 Patients in Korea

PONE-D-20-27008R1

Dear Dr. Lee,

We’re pleased to inform you that your manuscript has been judged scientifically suitable for publication and will be formally accepted for publication once it meets all outstanding technical requirements.

Kind regards,

Giuseppe Vergaro, M.D., Ph.D.

Academic Editor

PLOS ONE

Reviewer's Responses to Questions

**Comments to the Author**

1. If the authors have adequately addressed your comments raised in a previous round of review and you feel that this manuscript is now acceptable for publication, you may indicate that here to bypass the “Comments to the Author” section, enter your conflict of interest statement in the “Confidential to Editor” section, and submit your "Accept" recommendation.

Reviewer #1: All comments have been addressed

Reviewer #2: All comments have been addressed

2. Is the manuscript technically sound, and do the data support the conclusions?

Reviewer #1: Yes

Reviewer #2: Yes

3. Has the statistical analysis been performed appropriately and rigorously? 

Reviewer #1: Yes

Reviewer #2: Yes

4. Have the authors made all data underlying the findings in their manuscript fully available?

Reviewer #1: Yes

Reviewer #2: Yes

5. Is the manuscript presented in an intelligible fashion and written in standard English?

Reviewer #1: Yes

Reviewer #2: Yes

6. Review Comments to the Author

Reviewer #1: I'm Satisfied with the revision made by authors. The papaer has remarkably improved and is now suitable for publication. Well-Done!

Reviewer #2: Congratulations for the improvements. The only point is in the abstract: although authors changes the conclusion in the abstract of the main file, they did not change the conclusion of the abstract that authors include during the submission process.

7. PLOS authors have the option to publish the peer review history of their article (what does this mean?). If published, this will include your full peer review and any attached files.

Reviewer #1: No

Reviewer #2: **Yes: **Flavio A. Cadegiani, MD, MSc, Ph.D.

---

## [Editor Report · Acceptance letter]

1 Mar 2021

PONE-D-20-27008R1 

Effect of Renin-Angiotensin-Aldosterone System Inhibitors on Covid-19 Patients in Korea 

Dear Dr. Lee:

I'm pleased to inform you that your manuscript has been deemed suitable for publication in PLOS ONE. Congratulations! Your manuscript is now with our production department. 

Kind regards, 

on behalf of

Dr. Yoshihiro Fukumoto 

Academic Editor

PLOS ONE